# Use of an Intramedullary Allogenic Fibular Strut Bone and Lateral Locking Plate for Distal Femoral Fracture with Supracondylar Comminution in Patients over 50 Years of Age

**DOI:** 10.3390/medicina59010009

**Published:** 2022-12-20

**Authors:** Wen-Chin Su, Tzai-Chiu Yu, Cheng-Huan Peng, Kuan-Lin Liu, Wen-Tien Wu, Ing-Ho Chen, Jen-Hung Wang, Kuang-Ting Yeh

**Affiliations:** 1Department of Orthopedics, Hualien Tzu Chi Hospital, Buddhist Tzu Chi Medical Foundation, Hualien 970473, Taiwan; 2School of Medicine, Tzu Chi University, Hualien 970374, Taiwan; 3Institute of Medical Sciences, Tzu Chi University, Hualien 970374, Taiwan; 4Department of Medical Research, Hualien Tzu Chi Hospital, Buddhist Tzu Chi Medical Foundation, Hualien 970473, Taiwan; 5Graduate Institute of Clinical Pharmacy, Tzu Chi University, Hualien 970374, Taiwan

**Keywords:** distal femoral comminuted fracture, intramedullary allogenic fibular strut bone, knee society score, visual analogue scale for knee pain

## Abstract

*Background and Objectives*: Distal femoral fracture is a severe injury that makes surgery challenging, particularly comminuted fractures in the supracondylar region. This study aimed to evaluate the outcomes of distal femoral fracture treated with the application of an intramedullary fibular allogenic bone strut in open reduction and internal fixation (ORIF) with precontoured locking plates in patients over 50 years of age. *Materials and Methods*: The study retrospectively enrolled 202 patients over 50 years of age with traumatic comminuted distal femoral fracture (AO/OTA 33-A3, 33-C2 and 33-C3) treated with ORIF with a locking plate from January 2016 to December 2019. The two groups were divided into patients who received an intramedullary allogenic bone strut and those who did not. Patients were followed for at least 1 year, with their function scores and radiographic data recorded. *Results*: A total of 124 patients were recruited, comprising 60 men and 64 women with an average age of 62.4 ± 8.5 years. The 36 patients who had received an intramedullary allogenic fibular bone strut reported lower postoperative pain scores at 1 month and lower postoperative Knee Society Scores (KSS) at 3 months than the control group. The application of an intramedullary allogenic fibular bone strut appeared to be significantly correlated with better 3-month postoperative KSS. *Conclusions*: The ORIF of distal femoral comminuted fracture with an intramedullary allogenic fibular bone strut can reduce pain and improve knee function in the early stages of postoperative rehabilitation and may reduce the time to union in patients over 50 years of age.

## 1. Introduction

Distal femoral fracture is a severe injury that is challenging to treat operatively. Less than 1% of all fracture patterns are accounted for in adult patients [1]. Incidents such as traffic accidents are the most common causes for such injuries in younger patients, whereas older adults or osteoporotic patients are likely to experience a higher rate of falls and knee contusions. This bimodal distribution introduces different treatment options along the distal femoral fracture [2]. Comminuted fracture, displaced fragments, and intra-articular involvement are often present. ORIF adequately realigns and reconstructs a smooth articular surface and provides immediate postoperative stability, fostering early rehabilitation [3]. Conservative treatment is indicated only for those unable to undergo surgery, such as patients with life-threatening conditions, medical comorbidities, or nondisplaced and stable fractures [4]. Precontoured locking compression plates are the most common implant type for distal femoral fracture; they provide better stability than dynamic condylar screws or angled blade plates and are thus ideal for osteoporotic or comminuted bone. Minimally invasive plate application can decrease damage to the fracture site’s vascular supply [5]. Severe comminuted fracture patterns, poor bone quality, inadequate stabilization, insufficient blood supply, and infection can increase the nonunion rate. Nonunion rates ranged between 0% and 10% despite application of the locking plate [6]. Peschiera et al. noted that metaphyseal comminution, bone loss, and malalignment may contribute to a high nonunion rate and proposed that an allograft bone strut be considered when a medial cortical defect more than 2 cm in length is observed intraoperatively [7]. The use of allogenic bone graft has decreased due to challenges in allogenic bone graft procurement and the increased frequency of locking plate application; nevertheless, this method has been applied for decades. However, a biomechanical study has indicated that the use of intramedullary allogenic bone strut combined with locking plate provides superior mechanical stability in unstable osteoporotic proximal humeral fractures [8]. Suh et al. also documented that hybrid use of allogenic bone graft can provide global stability in total knee arthroplasty (TKA) supracondylar periprosthetic fracture [9].

Research has not explored the hybrid use of locking plates and allogenic bone strut in fresh distal femoral fracture. We conducted a single-center retrospective study to evaluate the perioperative trauma and surgical parameters, the functional outcomes as knee society score (KSS) and visual analogue scale (VAS) for knee pain, and the radiographic union status as radiographic union score of the femur (RUSF) of precontoured locking plates combined with allogenic bone strut use in comminuted distal femoral fracture.

## 2. Materials and Methods

The study was conducted in accordance with the Declaration of Helsinki and approved by the Research Ethics Committee of Hualien Tzu Chi Hospital, Buddhist Tzu Chi Medical Foundation. Our trauma center cares for 500,000 people in eastern Taiwan. This retrospective study enrolled 202 patients with new distal femoral fracture between January 2016 and December 2019 and followed the participants for at least 1 year. Eligibility criteria were as follows: patients over 50 years of age with a new traumatic comminuted distal femoral fracture (AO/OTA 33-A3, 33-C2 and 33-C3) treated with ORIF with a locking plate. An intramedullary allogenic bone strut application was determined mainly by two factors: (1) the preference and experience of the surgeon individually and (2) the existence or absence of allogenic bone strut in the bone bank at the time point of surgery. The exclusion criteria were as follows: patients with pathologic or concomitant fractures, active malignancy, or infection. The traumatic mechanism of injury and preoperative and postoperative VAS scores were recorded. Outcome measurements included postoperative radiological union and knee function with KSS. Adequate radiologic bone union was defined as the detection of callus bridging in three out of four cortices at the fracture site as observed through anterolateral and lateral radiography. We also used RUSF for evaluation of callus formation of the four cortices of the follow-up radiographs of the patients. RUSF was based on the assessment of healing at each cortex (i.e., medial and lateral cortices on the anteroposterior plain film as well as anterior and posterior cortices on the lateral film) [10].

### 2.1. Surgical Technique

The objectives of distal femoral fracture management are to achieve anatomic reduction of the joint surface and restore the limb’s length, rotation, and mechanical axis. The patient was positioned in the supine position with a cushion placed under the ipsilateral buttock. The original lateral approach was administered, and an incision was made in the iliotibial band parallel to the fiber. Vastus lateralis muscle fascia was incised and retracted anteromedially. Care was taken to avoid excess periosteal stripping. The Swashbuckler approach was used in the case of intra-articular comminuted fracture [11]. The surgical incision was extended in the lateral parapatellar approach, and the capsulotomy technique was performed upon approaching the articular surface. Fracture fragments should first be identified through radiography or computed tomography. Adequate reduction of large bone fragments should be completed with the fixation of interfragmentary screws, wire, or Kirschner wire (K-wire). In the presence of severe comminution, an intramedullary allogenic bone strut can be inserted through the gap between comminuted fragments. Indirect reduction of the medial comminuted fragment or oblique medial cortex is then achieved. Further medialization of the bone strut by the K-wire “joystick” manipulation technique can reduce the medial comminuted cortex to its original position and augment the medial column in the case of massive medial cortical bone defect (Figure 1). A bone strut can sometimes be inserted through the intercondylar notch [12]. If the comminuted condyle is too fragile to be reduced in size, the bone strut can be advanced distally for the indirect reduction of the distal femoral articular block and facilitation of distal screw purchase. Temporary fixation with K-wire and a reduction clamp can be used to align the mechanical axis and lateral distal femoral angle. If the limb length, rotation, and mechanical axis are restored, the precontoured locking plate can be applied to neutralize the fracture (Figure 2). We presented short- and long-term radiographic follow-up of a 49-year-old female patient (Figure 3) and a 19 -year-old male patient (Figure 4), and they both had good postoperative function recovery and bone union.

### 2.2. Source, Preparation and Storage of the Allogenic Bone Strut

The allogenic fibular strut bone was harvested from a brain-dead donor diagnosed by two different doctors. The donor was screened for syphilis (STS-RPR), HIV antibody (EIA), hepatitis B (HBs antigen), hepatitis C (anti-HCV antibody) and blood culture pre-operatively. If all the laboratory screening tests were normal, allograft harvest could be administered. After the allogenic bone strut was retrieved and cleaned by normal saline solution, bacteria culture was swabbed immediately on each bone graft. The bone graft was packed in three layers of sterile plastic bags and stored in the bone bank at −70 °C. The bone graft could be applied to the orthopedic surgery after all the intraoperative bacteria culture data were negative for bacterial growth.

### 2.3. Statistical Analysis

Statistical analysis was performed using SPSS for Windows, version 23.0 (IBM, Armonk, NY, USA). Descriptive statistics (means, standard deviations, ranges, coefficients of variation, and proportions) were calculated, and an independent t-test was used for comparisons. A generalized linear model (GLM) was used to evaluate risk factors associated with KSS at 3 months and 1 year postoperative.

## 3. Results

A total of 202 patients were enrolled in the study between January 2016 and December 2019. Four patients expired during hospitalization due to comorbidities. An additional 74 patients did not engage in regular follow-up care, and their data were thus incomplete. The remaining 124 patients comprised 60 men and 64 women with an average age of 62.4 ± 8.5 years (Table 1).

Traffic accidents were the cause of fracture for 80 patients. In total, 37 patients had AO/OTA type A3 fracture and the other 87 had AO/OTA type C1-3 fracture. The mean intraoperative blood loss was 580.3 ± 135.1 mL, and the mean length of hospital stay was 12.1 ± 10.0 days. We divided the patients into two groups based on their use or nonuse of an intramedullary allogenic fibular bone strut. Eighty-four patients received ORIF without intramedullary allogenic fibular bone strut and 40 patients received ORIF with intramedullary allogenic fibular bone strut (Table 1). In the postoperative evaluation of the distal femoral fracture, postoperative 1-month VAS (*p* = 0.043) score and postoperative 3-month KSS were significantly lower in the group that received the intramedullary allogenic fibular bone strut (*p* < 0.001; Table 2). 

Postoperative 3-month RUSFs were significantly better in the group that received the intramedullary allogenic fibular bone strut (*p* = 0.021; Table 2), while postoperative 1-year RUSFs were slightly better in the group that received the intramedullary allogenic fibular bone strut with marginal significance (*p* = 0.064; Table 2).The mean bone union period was 7.4 ± 2.2 months with no significant difference between groups (Table 2). We performed a risk analysis between postoperative 3-month KSS and postoperative 1-year KSS. According to the GLM results, the use of intramedullary allogenic fibular bone strut is significantly correlated with better postoperative 3-month KSS (*p* < 0.001); old age and male sex are significantly correlated with poorer postoperative 1-year KSS (*p* = 0.007 and 0.009; Table 3).

## 4. Discussion

Minimally invasive osteosynthesis is the current preferred distal femoral fracture treatment strategy. The previous technique, which entailed substantial stripping of the periosteum and destruction of surrounding soft tissue, can disrupt the vascular supply, contributing to delayed union or nonunion [13,14]. Despite the frequent application of minimally invasive plate osteosynthesis, nonunion rates of distal femoral fracture remain at 0–10% [15]. Researchers have demonstrated that the predisposal of fresh distal femoral fracture to nonunion is due to metaphyseal bone defects, an inability to obtain adequate bony fixation, and a failure to augment bone grafts to address metaphyseal comminution [16]. Kubiak et al. revealed that rigid fixation by locking plates may restrict fracture healing under the principle of secondary healing [17]. As reported by Peschiera et al., malreduction caused by axial defects and medial cortical bone defects are the major risk factors of nonunion [7]. Patients in whom these two problems were unaddressed were reported to have a nonunion rate of approximately 12%. Peschiera et al. also proposed the application of medial support, such as a bone strut allograft or medial buttress plate, for medial cortical defects over 2 cm in length.

The use of an intramedullary allogenic bone strut can resolve the aforementioned causes of nonunion. First, an intramedullary allogenic bone strut can reduce the rate of malreduction. It is difficult to align the comminuted distal femoral fracture using a minimally invasive technique. If a long allogenic bone strut is inserted into the diaphysis and metaphysis, the strut can realign and reduce the displaced and comminuted fragments. Second, bone defects caused by metaphyseal comminution can be corrected with additional bone graft struts. Poor screw purchase may be encountered when the locking screw is applied at a comminuted metaphysis and condyle. Better screw purchase can be obtained between the locking plate and bone strut of this loose area and can provide augmented fixation and early stability. Third, intramedullary allogenic bone struts can function as a substitute for medial cortical bone defects and provide additional screw purchase stability. The use of a medial buttress plate for this defect can help prevent periosteal stripping at the medial distal femur.

According to a biomechanical study on TKA periprosthetic distal femoral fracture by Chen et al., locking plate fixation with intramedullary allograft provided better construct stiffness and less fracture micromotion and implant stress than the use of a locking plate alone. An allogenic bone strut can aid in partial knee load transmission and decrease the moment arm between the allograft and condyles, which can reduce the mechanical demands of the lateral less invasive stabilization system and help stabilize osteosynthesis [18].

Concerns may arise over the substantial stripping of the periosteum and destruction to surrounding soft tissue during insertion of the intramedullary allogenic bone strut. Because we addressed the problems of metaphyseal bone defects, an inability to obtain adequate bony fixation, and a failure to augment bone grafts in cases of metaphyseal comminution, cases of nonunion were absent in our data set. Although the application of an intramedullary allogenic bone strut had no significant associations, the data indicated that application may decrease union time. In addition, both short-term VAS and KSS were found to have statistically significant relationships in the intramedullary allogenic bone strut group. According to the results, excellent biochemical stability is produced by better construct stiffness and less fracture micromotion and implant stress, and greater relief of postoperative pain and early rehabilitation and range of motion can be achieved. Older adults can anticipate a better prognosis and less postoperative comorbidity due to timely rehabilitation. In addition, a lower ORIF revision rate and reduction in social and financial burden can be expected.

This retrospective study has several limitations. First, the sample was small and nonrandomized. Second, data on comprehensive comorbidities with the potential to influence fusion time, such as diabetes mellitus and smoking history, were not recorded. We also did not evaluate the local bone density status of the knee of the patients. In addition, the small sample size did not allow us to evaluate the AO classification subtype. The distal femur bone stock could not be classified precisely because some patients did not undergo preoperative computed tomography. 

Nonetheless, the study demonstrated the efficacy of the intramedullary allogenic fibular bone strut among patients over 50 years of age with distal femoral fracture with comminution of the supracondylar region, especially in the early recovery stage. Future studies will focus on the comparison of this structure with other kinds of ORIF structures.

## 5. Conclusions

ORIF of comminuted distal femoral fracture with intramedullary allogenic bone fibular strut can reduce pain and improve knee function in the early stages of postoperative rehabilitation and may reduce union time. We particularly recommend intramedullary allogenic bone strut application for older patients.

## Figures and Tables

**Figure 1 medicina-59-00009-f001:**
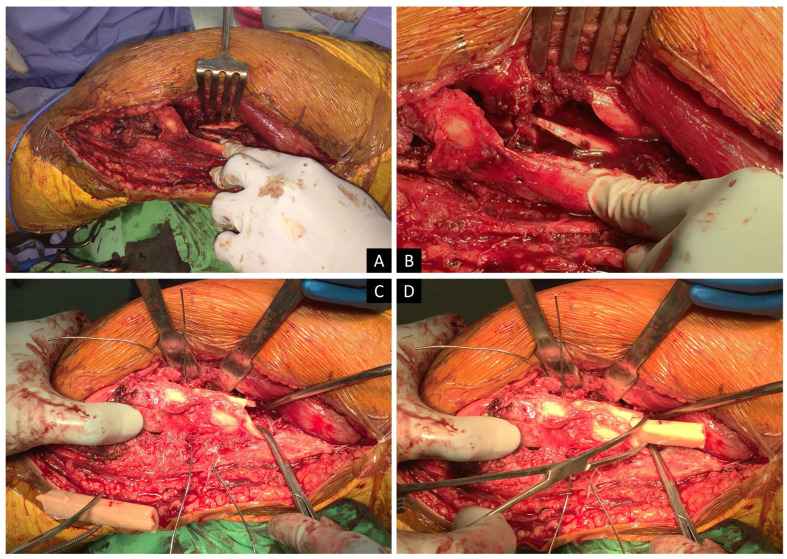
This is a 49-year-old female patient without any systemic disease admitted to our emergency department due to a traffic accident. Distal femoral fracture with AO33-C2 was diagnosed (Figure 3). Severe comminution of metaphysis with large bone defect was noted (**A**,**B**). We repositioned all the bone fragments (**C**) and an allogenic fibular strut was chosen for restoration of the fractured bone. Wire was applied as an outer restriction for fragment reposition (**D**). Allogenic bone strut supplied an inner supportive structure and wire was tied. Last, a precontoured locking plate was applied and proximal screws were inserted with minimal invasive technique.

**Figure 2 medicina-59-00009-f002:**
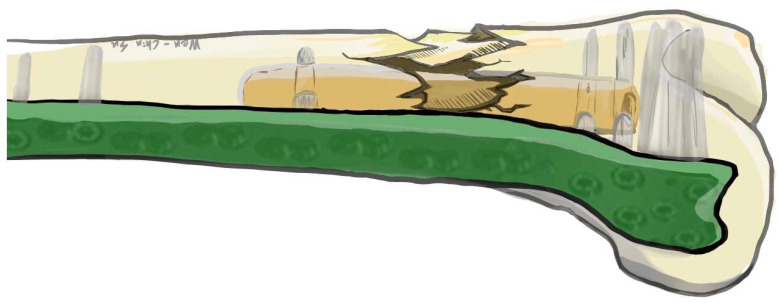
This figure shows the intramedullary application of the strut bone for distal femoral fracture. The strut bone provides fixation stability and medial cortex support.

**Figure 3 medicina-59-00009-f003:**
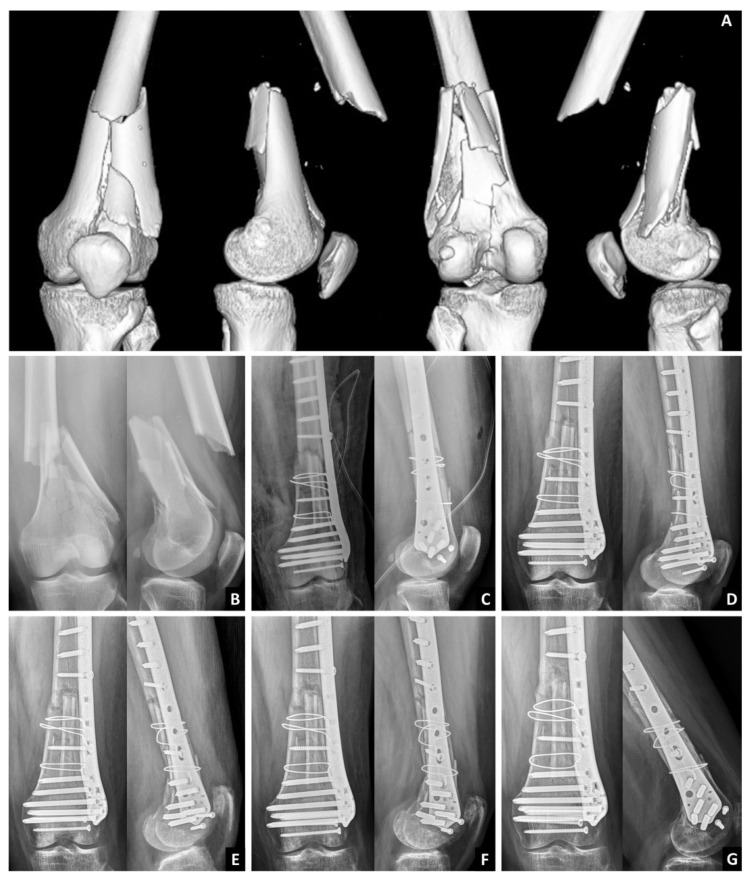
Radiography and computed tomography of the 49-year-old female patient with comminuted distal femoral fracture and good surgical result. (**A**) CT; (**B**) Pre-OP; (**C**) Post-OP; (**D**) Post-OP 1 month; (**E**) Post-OP 3 months; (**F**) Post-OP 6 months; (**G**) Post-OP 1 year.

**Figure 4 medicina-59-00009-f004:**
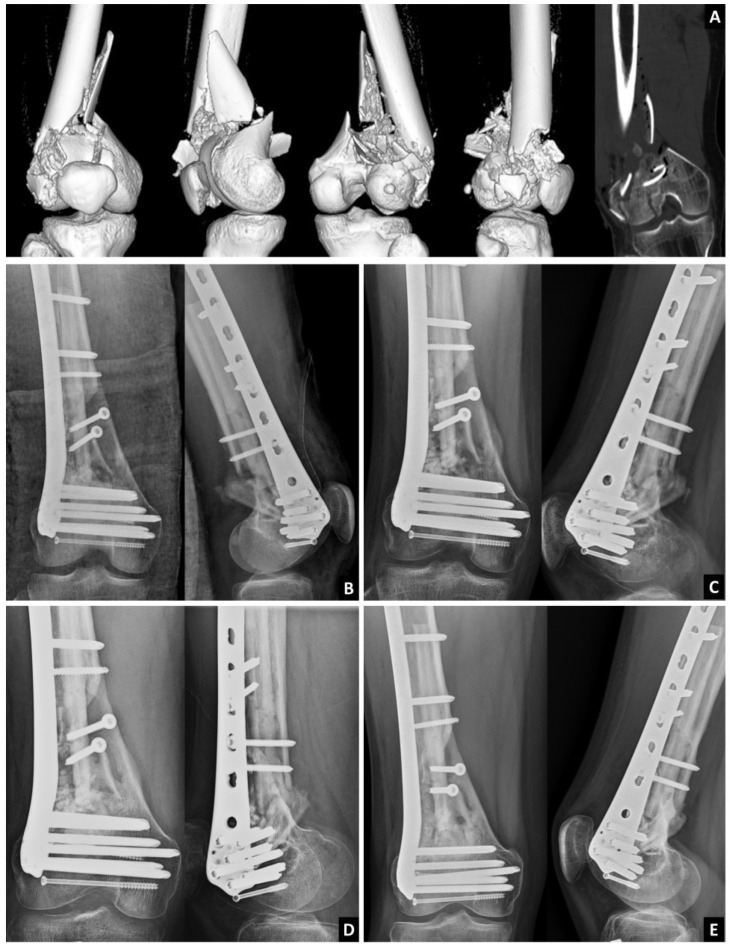
A 19-year-old male patient was involved in a traffic accident. Distal femoral open fracture with a 4 cm open wound at the anterolateral side of the distal femur. Computed tomography showed a severe comminuted metaphysis, AO 33-C2. After adequate debridement and irrigation, an allogenic fibular strut was applied. Medial cortex was fixed on the fibular strut. Allogenic TKA bone chip was stuff in the bone defect. (**A**) CT; (**B**) Post-OP; (**C**) Post-OP 3 months; (**D**) Post-OP 6 months; (**E**) Post-OP 1 year.

**Table 1 medicina-59-00009-t001:** Demographic data of the included patients (*n* = 124).

Variable	Without Fibular Strut	With Fibular Strut	Total	*p*-Value
N	84	40	124	
Age	61.3 ± 9.7	64.9 ± 9.9	62.4 ± 8.5	0.065
Gender	-	-	-	0.082
Male	38 (45.2%)	14 (35.0%)	52 (41.9%)	
Female	46 (54.8%)	26 (65.0%)	72 (58.1%)	
Mechanism	-	-	-	0.073
Fall from height	26 (31.0%)	18 (45.0%)	44 (35.5%)	
Traffic accident	58 (69.0%)	22 (55.0%)	80 (64.5%)	
AO Type	-	-	-	0.086
A3	21 (25.0%)	16 (40.0%)	37 (29.8%)	
C	63 (75.0%)	24 (60.0%)	87 (70.2%)	
Blood loss	666.4 ± 152.5	475.6 ± 92.8	580.3 ± 135.1	0.102
Length of Stay	15.9 ± 5.8	10.0 ± 2.2	12.1 ± 10.0	0.144

Data are presented as *n* or mean ± standard deviation.

**Table 2 medicina-59-00009-t002:** Postoperative functional evaluation of both groups (*n* = 124).

Variable	Without Fibular Strut	With Fibular Strut	Total	*p*-Value
VAS for knee pain (2W)	4.9 ± 1.1	5.1 ± 0.7	4.9 ± 0.9	0.328
VAS for knee pain (1M)	2.5 ± 1.0	2.0 ± 0.9	2.3 ± 0.9	0.043 *
Knee society score (3M)	66.4 ± 3.6	77.7 ± 1.5	69.7 ± 6.1	<0.001 *
Knee society score (1Y)	84.4 ± 5.0	85.8 ± 3.2	84.8 ± 4.6	0.209
Radiographic union score of the femur (3M)	6.4 ± 2.3	8.3 ± 2.2	7.0 ± 2.1	0.021 *
Radiographic union score of the femur (1Y)	9.5 ± 1.8	10.3 ± 1.3	9.8 ± 1.5	0.064
Union period (M)	7.7 ± 2.5	6.9 ± 0.9	7.4 ± 2.2	0.105

Data are presented as *n* or mean ± standard deviation. * *p*-value < 0.05 was considered statistically significant after test. VAS: visual analogue scale; M: month; Y: year.

**Table 3 medicina-59-00009-t003:** Factors associated with knee society score at 3 months and 1year after operation among patients (*n* = 124).

Variable	Knee Society Score (3M)		Knee society Score (1Y)	
	*β* (95% CI)	*p*-Value	*β* (95% CI)	*p*-Value
Intercept	65.76 (62.32, 69.21)	<0.001 *	88.69 (83.99, 93.39)	<0.001 *
Age	−0.02 (−0.06, 0.02)	0.365	−0.08 (−0.14, −0.02)	0.007 *
Gender	-	-	-	-
Male	1.15 (−0.32, 2.62)	0.123	−2.69 (−4.70, −0.68)	0.009 *
Female	References	NA	References	NA
Mechanism	-	-	-	-
Fall down	0.38 (−1.19, 1.95)	0.630	2.00 (−0.13, 4.14)	0.066
Traffic accident	References	NA	References	NA
LOS	0.05 (−0.02, 0.13)	0.129	−0.02 (−0.12, 0.08)	0.665
AO Type	-	-	-	-
A3	References	NA	References	NA
C	−0.25 (−1.92, 1.43)	0.770	0.13 (−2.16, 2.42)	0.912
Locking Plate	-	-	-	-
No	References	NA	References	NA
Yes	0.36 (−1.54, 2.25)	0.709	1.02 (−1.57, 3.60)	0.436
Application of fibular strut graft	-	-	-	-
No	References	NA	References	NA
Yes	12.04 (10.46, 13.63)	<0.001 *	1.04 (−1.12, 3.20)	0.341

Data are presented as odds ratio (95% CI). * *p*-value < 0.05 was considered statistically significant after test. M: month; Y: year.

## Data Availability

Data are contained within the article.

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
