# Peer review of "Use of an Intramedullary Allogenic Fibular Strut Bone and Lateral Locking Plate for Distal Femoral Fracture with Supracondylar Comminution in Patients over 50 Years of Age"

_medicina, 2022, doi:10.3390/medicina59010009_

Round 1
Reviewer 1 Report
The work focuses on a clinical study carried out in order to evaluate the effect of the use of an intramedullary bone strut for distal femur fractures. I only have a couple of comments:
- in the introduction it could be of help if you mention the parameters such as KSI you use later in the owrk
- why didn't you evaluate the bone density, or more in general callus formation, as a parameter for comparison?
- it is not fully clear to me if the compariosn between the treatments holds given that the starting situation of the treated patients was comaprable. Please mention this as it affects the comaprison and therefore the results.
Author Response
Reply to the Review Report (Reviewer 1)
Comments and Suggestions for Authors
The work focuses on a clinical study carried out in order to evaluate the effect of the use of an intramedullary bone strut for distal femur fractures. I only have a couple of comments:
- in the introduction it could be of help if you mention the parameters such as KSS you use later in the work
Ans: We thank your time and effort to enhance the quality of our work. We have supplemented the last paragraph of introduction as below: “Research has not explored the hybrid use of locking plates and allogenic bone strut in fresh distal femoral fracture. We conducted a single-center retrospective study to evaluate the perioperative trauma and surgical parameters, the functional outcomes as knee society score (KSS) and visual analogue scale (VAS) for knee pain, and the radiographic union status as radiographic union score of the femur (RUSF) of precontoured locking plates combined with allogenic bone strut use in comminuted distal femoral fracture.”
- why didn't you evaluate the bone density, or more in general callus formation, as a parameter for comparison?
Ans: Thank you for your reminding. We have added the estimation of bone density into the limitation section in Discussion as below:” We also did not evaluate the local bone density status of the knee of the patients.“ We also added the callus evaluation scores into our study as below: “We also used RUSF for evaluation of callus formation of the 4 cortices of the follow-up radiographs of the patients. RUSF was based on the assessment of healing at each cortex; i.e., medial and lateral cortices on the anteroposterior plain film as well as anterior and posterior cortices on the lateral film” The results of bone union scale as RUSF were added into table 2 and the Result section as below:” Postoperative 3-month RUSF were significantly better in the group that received the intramedullary allogenic fibular bone strut (p = 0.021; Table 2), while postoperative 1-year RUSF were slightly better in the group that received the intramedullary allogenic fibular bone strut with marginal significance (p = 0.064; Table 2).”
- it is not fully clear to me if the comparison between the treatments holds given that the starting situation of the treated patients was comparable. Please mention this as it affects the comparison and therefore the results.
Ans: Thank you for your reminding. We have added the p-value in table 1 as a comparison between the demographic data of the two groups. According to our revised table 1, there were no significant difference between the demographic parameters of the two groups. It means that their results were comparable. We have added this description into Result section as below: “There were no significant differences between all the demographic data of both groups”.

Reviewer 2 Report
In this manuscript, the authors reports the results of retrospective study comparing the outcome of two different surgical technique used for distal femur fracture with supracondylar comminution in patients over 50 years of age: application of an intramedullary fibular allogenic bone strut and internal fixation with locking plates. The results suggests the superiority of application of allogenic strut bone since the patients have reduced pain and improved knee function earlier during postoperative rehabilitation.
The Abstract and Introduction are concise and well written and the main objective of the study is clear.
In the Material and methods section there are several questions that has to be briefly explained:
1. Line 77: „A surgeon determined the suitability of an intramedullary allogenic bone strut application.“ : please add a criteria according to which the surgeon(s) decides to insert an allogenic bone graft.
2. Very important is the source of bone allografts (Biobank, Bone bank or any other unit that care for bone transplantation surgery) and maybe more important is how these fibular allogenic bone struts are prepared and stored. That is very crucial for osteoconductive and osteoinductive properties of the allografts and therefore for the process of the fracture healing. Please, state in a few words: the source of allografts, the main technique of sterilization and storage of bone grafts (fresh frozen, gama radiation, pasteurization, chemically sterilized bone) .
3. In section Surgical technique, Figure 1, the mark (A), (B) (C) or (D) should be add in description of photographs, e.g. „Severe comminution of metaphysis with large bone defect was noted (A).“
4. Figure 3. and Figure 4. it would be better if you add „Post-OP 1 month“ instead „Post-OP 1m“.
5. Table 1. Demographic dana of the included patients: explain W/O Fibular strut and W/I Fibular strut (I believe that mean without and with allogenic bone) – in table you have 40 patients who received ORIF with intramedullary allogenic bone but in text you state that you have 36 patients (line 157) – please correct the number…
Author Response
Reply to the Review Report (Reviewer 2)
Comments and Suggestions for Authors
In this manuscript, the authors reports the results of retrospective study comparing the outcome of two different surgical technique used for distal femur fracture with supracondylar comminution in patients over 50 years of age: application of an intramedullary fibular allogenic bone strut and internal fixation with locking plates. The results suggests the superiority of application of allogenic strut bone since the patients have reduced pain and improved knee function earlier during postoperative rehabilitation.
The Abstract and Introduction are concise and well written and the main objective of the study is clear.
In the Material and methods section there are several questions that has to be briefly explained:
- Line 77: „A surgeon determined the suitability of an intramedullary allogenic bone strut application.“ : please add a criterion according to which the surgeon(s) decides to insert an allogenic bone graft.
Ans: Dear reviewer, thank you for your suggestion. We have added the content as below: ”An intramedullary allogenic bone strut application was determined mainly by two factors: (1) the preference and experience of the surgeon individually; (2) the existence or absence of allogenic bone strut in the bone bank at the timepoint of surgery.”
- Very important is the source of bone allografts (Biobank, Bone bank or any other unit that care for bone transplantation surgery) and maybe more important is how these fibular allogenic bone struts are prepared and stored. That is very crucial for osteoconductive and osteoinductive properties of the allografts and therefore for the process of the fracture healing. Please, state in a few words: the source of allografts, the main technique of sterilization and storage of bone grafts (fresh frozen, gamma radiation, pasteurization, chemically sterilized bone) .
Ans: Dear reviewer, we appreciate the reviewer for the positive feedback. We have added the source, preparation and storage of the allogenic bone strut in Materials and Methods as 2-2 as below: “The allogenic fibular strut bone was harvested from a brain death donor diagnosed by 2 different doctors. The donor was screened for syphilis (STS-RPR), HIV antibody (EIA), hepatitis B (HBs antigen), hepatitis C (Anti-HCV antibody) and blood culture pre-operatively. If all the laboratory screening tests were normal, allograft harvest could be administered. After the allogenic bone strut was retrieved and cleaned by normal saline solution, bacteria culture was swabbed immediately on each bone graft. The bone graft was packed in 3 layer of sterile plastic bags and storage in the bone bank at -70 °C. The bone can be applied to the orthopedic surgery after all the intra-operative bacteria culture data were all negative for bacterial growth.”
- In section Surgical technique, Figure 1, the mark (A), (B) (C) or (D) should be add in description of photographs, e.g., „Severe comminution of metaphysis with large bone defect was noted (A).“
Ans: Dear reviewer, thank you for your reminding. We have added the marks into the description of the figure.
- Figure 3. and Figure 4. it would be better if you add „Post-OP 1 month“ instead „Post-OP 1m“. Post-OP 1 month
Ans: Dear reviewer, thank you for your reminding. We have modified these mistakes.
- Table 1. Demographic dana of the included patients: explain W/O Fibular strut and W/I Fibular strut (I believe that mean without and with allogenic bone) – in table you have 40 patients who received ORIF with intramedullary allogenic bone but in text you state that you have 36 patients (line 157) – please correct the number.
Ans: Dear reviewer, thank you for your reminding. We have modified these mistakes.
